

# Use of different dry materials to control the moisture in a black soldier fly (*Hermetia illucens*) rearing substrate

Parichart Laksanawimol[1], Pritsana Anukun[1] and Anchana Thancharoen[2]

[1] Faculty of Science, Chandrakasem Rajabhat University, Bangkok, Thailand
[2] Department of Entomology, Faculty of Agriculture, Kasetsart University, Bangkok, Thailand

## ABSTRACT

**Background**. Controlling the substrate moisture is a significant challenge in black soldier fly (BSF) farming. Many substrates have a high moisture content, which results in a low BSF biomass and a high mortality. One potential solution involves incorporating dry substrates into the food mix to mitigate the excessive moisture. However, little information about the types and quantities of dry substrates is available.
**Methods**. Six different dry materials–rice husk (RH), rice bran (RB), rice husk ash (RHA), coconut coir dust (CC), rubberwood sawdust (RSD), and spent coffee grounds (SCGs)–were evaluated by combining with pure minced mixed vegetables in varying proportions (0%, 5%, 10%, 15%, 25%, and 50% by weight). This study encompassed both small-scale and medium-scale experiments to comprehensively assess the effects of the addition of each of these different dry substrates and their quantities on aspects of the development of BSF, such as BSF biomass, larval duration, mortality rates, adult sex ratio, and the moisture removal efficiency of each substrate mixture.
**Results**. Each dry substrate had specific properties. Although RB emerged as a favorable dry substrate owing to its nutritional content and substantial water-holding capacity, excessive use of RB (>15% by weight) resulted in elevated temperatures and subsequent desiccation of the substrate, potentially leading to larval mortality. In contrast, RH demonstrated the ability to support improved larval duration and growth, permitting its utilization in higher proportions (up to 50%). On the other hand, CC, RHA, and SCG are better suited for inclusion in BSF larval substrates in smaller quantities.
**Discussion**. Some dry substrates require a pretreatment process to eliminate toxic substances prior to their incorporation into substrate mixtures, such as CC and SCG. A potential alternative solution involves employing a combination of various dry substrates. This approach aims to enhance the substrate moisture control and subsequently improve the BSF rearing performance.

# INTRODUCTION

The black soldier fly (BSF), *Hermetia illucens* (L.) (Diptera: Stratiomyidae), has gained in popularity recently as a biological waste recycler, converting the waste into larval biomass that can then serve as a protein source for diet supplementation in poultry and livestock rearing (*da Silva & Hesselberg, 2020*).

Corresponding author
Anchana Thancharoen,
koybio@gmail.com

There have been no reports of BSF larvae (BSFL) having a negative effect on the growth performance or meat quality in pigs or poultry, although a lower feed conversion rate has been reported in pigs (*Lu et al., 2022*). The high amount of fat obtained from the defatting BSFL can be used to produce high-quality biodiesel because it contains high levels of lauric, palmitic, and myristic acids (*Kim et al., 2021*). In addition, the fat from BSFL has a high potential for use in cosmetics (*Franco et al., 2022*). The BSF frass, a byproduct from the larval decomposition process, is composed of 37% C, 3% N, 1–5% P, and 0.5–4.1% K by wet weight (w/w) and so can be utilized as a biofertilizer (*Lopes, Yong & Lalander, 2022*). As a result of these potential benefits from BSF rearing, insect farming is being practiced worldwide in an effort to create a sustainable circular bioeconomy by extracting value from biowaste management (*Kim et al., 2021*; *Liu et al., 2022*; *Lopes, Yong & Lalander, 2022*).

When it comes to BSF farming, there are several substrate factors that can affect the ability of BSFL to efficiently convert waste materials. These factors include the nutrient content, fiber content, particle size, substrate ventilation, pH, structure, and moisture content (*Barrett et al., 2023*; *Cheng, Chiu & Lo, 2017*; *Joly & Nikiema, 2019*; *Ma et al., 2018*; *Yakti et al., 2023*; *Zhang et al., 2021*). The nutrient content of the substrate plays a significant role in the growth, development, and body composition of the BSFL (*Barragán-Fonseca et al., 2018*; *Eggink et al., 2022*; *Meneguz et al., 2018*). While BSFL can survive on substrates with a wide range of pH levels, from highly acidic to highly basic (pH 2.0–10.0), the optimum initial pH for favorable gut microorganisms and higher biomass production ranges from pH 6.0 to 8.0 (*Ma et al., 2018*).

Substrate ventilation is crucial to allow oxygen diffusion for consumption by the BSFL. These insects prefer feeding on the top 2–4 cm of the substrate, avoiding the anaerobic zone towards the bottom (*Barrett et al., 2023*). Furthermore, the moisture content of the substrate influences the vertical distribution behavior of the BSFL. Indeed, BSFL are found at different substrate depths depending on the moisture content, with higher moisture contents leading to higher mortality rates (MRs) (*Bekker et al., 2021*). The moisture content significantly affects the BSF larval abilities and performances, with negative effects on feed reduction and wet weight and positive effects on developmental time. One study indicated that BSFL reared on a substrate with a 60% moisture content exhibited the highest growth performance, while the growth was lowest at an 80% moisture content (*Dzepe et al., 2020*). In contrast, a moisture content below 80% was found to have no effect on larval survival rates, while above 90% it reduced the survival rates (*Dzepe et al., 2020*; *Khairuddin, Ghafar & Hassan, 2022*; *Lalander et al., 2020*. However, the larvae were unable to develop on a 40% moisture content (*Cammack & Tomberlin, 2017*). In fact, the moisture contents did not influence BSFL performance directly but were directly associated with the activities of substrate microbiota (*Bekker et al., 2021*). In addition, high-moisture substrates can present challenges during residue separation at harvest time (*Cheng, Chiu & Lo, 2017*). By considering and optimizing these substrate factors, BSF farmers can enhance the overall efficiency and success of their farming practices while ensuring the well-being and productivity of the BSFL.

Many organic wastes, such as food waste, fruits and vegetables, have a high water content. Due to the importance of substrate moisture on biomass production, several techniques

have been evaluated to reduce high moisture levels and increase larval yield. One approach is the design of an automated incubation system that utilizes pressure sensors to control the temperature and maintain a substrate moisture level of 60% through water evaporation (*Erbland, Alyokhin & Peterson, 2021*). Another method involves increasing the aeration rate of the larval rearing containers, which can help control the air temperature and relative humidity, ultimately reducing the substrate moisture (*Abduh et al., 2022*; *Cheng, Chiu & Lo, 2017*). Additionally, excess moisture can be minimized by adding dry substrates, such as coconut coir dust (CC), palm kernel meal, or milk powder, on top of the substrate to adjust the water content to below 80% (*Dortmans, Diener & Verstappen, 2017*). Despite the use of some dry substrates, limited information is available regarding the optimum quantity of each type of dry substrate to be added.

The objective of this study was to evaluate the efficiency of various types of dry substrates when added to a minced mixture of vegetables with high moisture content. Each dry substrate, with its distinct physical properties, may exhibit a different efficiency in controlling excessive moisture and impact the BSFL performance differently. Therefore, six different dry substrates derived from agricultural byproducts were selected, and their quantities were varied to formulate suitable diets with optimal substrate moisture levels, in terms of enhancing the BSFL rearing process. These dry substrates offer a potential low-cost technique that can benefit BSF farming, especially for small-scale production where automated machinery may be lacking.

## MATERIALS & METHODS

### Maintenance of the BSF colony

The BSFs used in this study were derived from eggs obtained from a local farmer in Khon Kaen province in 2019. The colony has since been maintained and bred at the Department of Entomology, Kasetsart University, Bangkok, Thailand. The adult flies were housed in a mesh chamber (4 (L) ×3 (W) ×3 (H) m) to support mating in flight and oviposition behavior. The flies were provided with a 30% (w/v) brown sugar solution, which was sprayed onto the mesh chamber three times a day to serve as a food source. Oviposition devices, consisting of a tray filled with an attractant substrate made from a mixture of wheat bran, fruits, and vegetables (*Khaekratoke, Laksanawimol & Thancharoen, 2022*; *Laksanawimol, Singsa & Thancharoen, 2023*), were placed within the chamber. Wooden sheet pieces were also provided as oviposition substrates placed above the tray of attractant materials. The gravid BSF females laid their eggs in the small gaps between wooden sheets. The BSFL were fed a minced mixture of organically grown vegetable waste (cabbage, romaine, tomato, and lettuce) obtained from the Royal Project Foundation. The rearing of the BSFL took place under natural ambient conditions, with temperatures ranging from 28 to 34 °C, relative humidity ranging from 70% to 90%, and a light: dark cycle of 13:11 h.

### Preparation of BSF eggs and larvae for experiment

The synchronous development of the eggs was achieved by carefully removing them from the wooden sheets using a cutter one day prior to the start of the experiment. In order to promote rapid growth and obtain larger BSFL for easier handling and enumeration,

fish protein was incorporated into the neonate diet. Each 0.25 g of eggs was subsequently transferred to a hatching container containing a baby food mixture composed of 500 g of cornmeal, 400 g of minced mixed vegetables, 500 g of rice bran, and 200 g of fish protein (feed rate = 110 mg/larva/d). This diet formulation aimed to produce 5-d-old larvae (referred to as 5-DOL).

## Experimental design

In the experiment, six different dry materials, namely rice husk (RH), rice bran (RB), rice husk ash (RHA), coconut coir dust (CC), rubberwood sawdust (RSD), and spent coffee grounds (SCG), were evaluated. Prior to use, the SCG was allowed to self-ferment for one to two weeks (modified from (*Khaekratoke, Laksanawimol & Thancharoen, 2022*)). The standard diet or control group (CON) was comprised of minced organically grown mixed vegetables, comprised of cabbage, romaine lettuce, tomato, and lettuce and then mixed with 10% (w/w) cornmeal. Various quantities, as % (w/w) of the dry substrates (5%, 10%, 15%, 25% and 50%) were mixed with the standard diet to create the test larval food substrates. The feed trials were divided into two phases: a small-scale experiment and a medium-scale experiment. This division aimed to identify an effective dry substrate that could adjust the substrate moisture to an optimal range.

Small-scale experiments were conducted in 2-liter plastic bowls with dimensions of 18 cm in diameter and 10 cm in height. The bowls were covered with a fine cloth to protect against other insects and promote air circulation. The tested substrates were prepared for a duration of 14 d at a feed rate of 200 mg/larva/d (*Nyakeri et al., 2019*). The standard diet was mixed with varying amounts of the dry substrate. The total diet in each replicate amounted to 840 g. Four replicates, each containing 300 5-DOL with an average size of $0.045 \pm 0.009$ g/larva, were gently handled using soft forceps and released into the different treatment diets. Two trials were performed in the small-scale experiment to assess the types of dry substrates and their percentages by wet weight (w/w).

In Trial 1, referred to as the high quantity test, the following dry material percentages were tested: 0% (CON), 25%, and 50%.

In Trial 2, referred to as the low quantity test, the following dry material percentages were tested: 0%, 5%, 10%, and 15%.

In the medium-scale experiments, a plastic tray with dimensions of 60 cm (L) ×40 cm (W) ×15 cm (H) was used. The tray was covered with a fine cloth to evaluate the efficiency of moisture control. A total of 8 kg of substrate was placed in the tray. The dry substrates that showed the highest potential in the small-scale experiment were selected for the medium-scale experiments. The selected dry substrates and their respective quantities, all as (w/w), were as follows: 5% RHA, 5% RB, 10% RH, 15% SCG, and the CON group without any dry substrate.

In each treatment, the quantity of dry substrate was varied while maintaining a fixed feed rate of 200 mg/larva/d. Three replicates were set up for each tested substrate. A small cup containing 0.24–0.29 g of BSF eggs (equivalent to 0.5 g or approximately 17,330 eggs was placed on each substrate and allowed to hatch.

## Developmental duration, growth performance, and survival rate of BSFL

To determine the growth performance of BSFL in different substrate conditions, 40 larvae, prepupae, and pupae were randomly selected from all replicates. These individuals were carefully cleaned and weighed using an analytical balance (OHAUS Pioneer PA214 Analytical Balance, Parsippany, NJ, USA). For the larval stage, samples were collected when they reached 40% prepupal appearance and had a creamy-white color. The prepupae, which exhibited a black larval cuticle, were collected at 80% prepupal appearance. After data collection, the sampled BSFL were returned to their respective experimental containers.

At the onset of prepupal appearance, the daily count of prepupae was recorded, and they were transferred to pupation containers to observe the developmental duration during the larval stage. The total larval duration, including the 5-d period of feeding baby food, was calculated.

In the small-scale experiments, the mortality rate (MR) was determined by counting the number of prepupae that successfully developed in each replicate. The MR was calculated using the following equation:

$$MR(\%) = [(\text{number of 5-DOL} - \text{total number of prepupae})/\text{number of 5-DOL}] \times 100 \quad (1)$$

## The sex ratio of BSF adults

After the BSFL reached the prepupal stage, they were transferred to 32-ounce plastic cups with aerated lids and provided with RSD as a pupation substrate. Once the adult BSF emerged, their sex was determined through visual inspection. This allowed for the identification and calculation of the percentage of females in the adult stage.

## Abiotic factor measurement

The temperature (°C) and relative humidity (%) inside the rearing room were continuously monitored using a datalogger (Elitech RC-51H, London, UK) to assess the abiotic factors that could affect the substrate moisture. Hourly measurements were recorded throughout the experiment. The moisture content (%) of all the substrates was measured daily using a soil moisture meter (Landtek MC-7828SOIL, China) from the 5-DOL stage until the prepupal stage. To evaluate the efficiency of moisture removal, the percentage of moisture removal (% M) was calculated by comparing the average moisture content in the CON treatment with the minimum moisture content at the end of each substrate experiment. The calculation (modified from (*Yakti et al., 2023*)) was performed using the following equation:

$$M(\%) = [(\text{moisture in CON} - \text{min moisture in substrate test})/\text{moisture in CON}]$$
$$\times 100. \quad (2)$$

## Data analysis

The data obtained for the mean weight of BSF (larvae, prepupae, and pupae), larval developmental time, percentage of female adults, larval survival rate, and substrate moisture content in the different substrate groups were subjected to statistical analysis. One-way

analysis of variance (ANOVA) was conducted to determine if there were significant differences among the substrate groups. For multiple comparisons, Fisher's least significant difference (LSD) test was performed. Independent sample $t$-tests were used to assess the significance of differences between specific treatments. Correlations between the efficiency of moisture removal and larval duration were determined by Pearson correlation at 95% confidence levels ($P < 0.05$). The statistical software SPSS, version 14 (SPSS for Windows, Chicago, IL, USA), was utilized for all statistical analyses. A significance level of $P < 0.05$ was considered statistically significant.

# RESULTS

## Larval duration

In Trial 1, when high levels of dry substrates were added, the BSFL exhibited a longer larval duration except for the substrates with 25% RB, 25% RH, or 25% SCG that exhibited a shorter larval duration compared to the CON ($11.9 \pm 1.6$ d, $12.6 \pm 1.7$ d, and $13.3 \pm 2.5$ d, respectively, *versus* $13.7 \pm 2.3$ d; $F = 500.844$, $df = 12$, $P = 0.000$) (Fig. 1A). In addition, most of the media with 50% (w/w) dry substrates resulted in a longer larval duration, except for the 50% RH, which had a shorter harvest time compared to the control ($13.5 \pm 2.2$ d) (Fig. 1A).

In Trial 2, where various low quantities of dry substrates were used, the BSFL generally had shorter larval durations, except for the 10–15% SCG ($F = 587.937$, $df = 18$, $P = 0.000$) (Fig. 1B).

## Larval survival

In contrast to the CON group ($6.7 \pm 4.1\%$ MR), a higher MR was observed in larval stages when high quantities (up to 25%) of dry substrates were present. Specifically, substrates containing, all % (w/w), 25–50% RB, 50% SCG, and 50% RHA showed elevated MRs ($17.8 \pm 10.0\%$, $25.3 \pm 5.8\%$, $22.7 \pm 7.1\%$, and $19.2 \pm 5.7\%$, respectively; $F = 8.362$, $df = 12$, $P = 0.000$) (Fig. 2A). However, when 25% (w/w) dry substrates were used, the MR was low, except for with RB due to the absence of moisture in this substrate.

In Trial 2, high MRs were observed when the larval food was mixed with all % (w/w), 15% RHA, 10–15% SCG, and 5–10% CC ($12.8 \pm 6.1\%$, $18.3 \pm 8.3\%$, $13.2 \pm 8.3\%$, $19.8 \pm 6.8\%$ and $15.8 \pm 5.7\%$, respectively; $F = 4.723$, $df = 18$, $P = 0.000$). Among the low dry substrate addition mixtures, only the CC and SCG treatments exhibited a high MR.

## Growth performance

Due to the consistent feed rate supply, mixing with higher quantities of dry substrates resulted in a lower substrate nutrition level and reduced biomass in the BSF larvae prepupae and pupae (Fig. 3). Compared to the control substrate, a wet weight loss of over 40% was observed in the larvae, prepupae, and pupae when cultured with media containing 50% (w/w) of any of the dry substrate mixtures except for the 50% RH ($F = 109.213$, $df = 12$, $P = 0.000$; $F = 72.886$, $df = 12$, $P = 0.000$ and $F = 139.152$, $df = 12$, $P = 0.000$, for the larvae, prepupae, and pupae, respectively). Interestingly, BSF reared on substrate mixtures containing 25–50% (w/w) RH displayed the highest weight compared to those with the
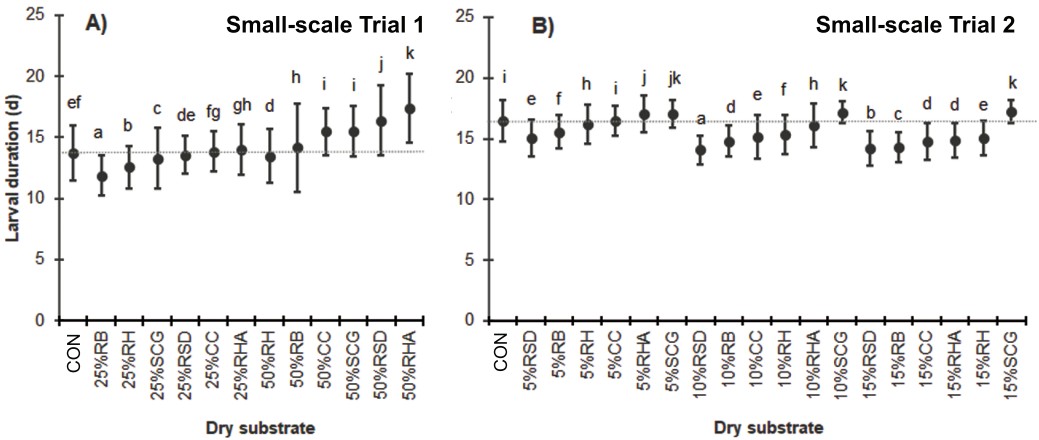

**Figure 1** Larval duration (d) of BSFL reared on various dry substrates in small-scale trials (A) 1 and (B) 2. Data are shown as the mean ± 1 SD, derived from four replicates. Different letters indicate significant differences among substrate treatments ($P < 0.05$).

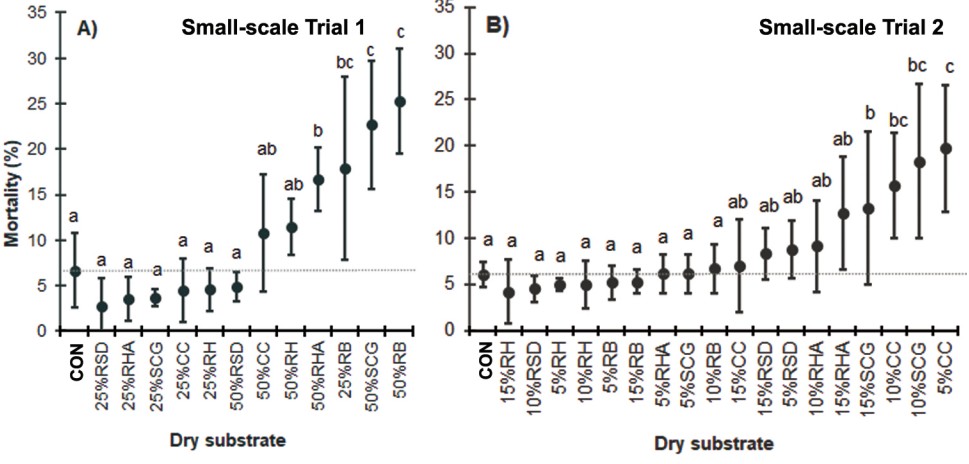

**Figure 2** The MR (%) of BSFL reared on various dry substrate trials in small-scale trials (A) 1 and (B) 2. Data are shown as the mean ± 1 SD, derived from four replicates. Different letters indicate significant differences among substrate treatments ($P < 0.05$).

same quantity of the other dry substrates. In contrast, 25–50% (w/w) RHA resulted in the smallest BSFL biomass, although the addition of a lower quantity of 10–15% (w/w) RHA showed improved results. Similarly, CC treatments showed the smallest biomass at the lower levels of added dry substrates (5–15% (w/w)), with a weight loss of 50–61% in the larval stage ($F = 97.158$, $df = 18$, $P = 0.000$ in larvae; $F = 140.912$, $df = 18$, $P = 0.000$ in prepupae, and $F = 82.832$, $df = 12$, $P = 0.000$ in pupae).

## Sex ratio

Overall, the proportion of female offspring produced ranged from 48% to 51%. The sex ratio of the offspring was not significantly affected by the addition of a low level of each

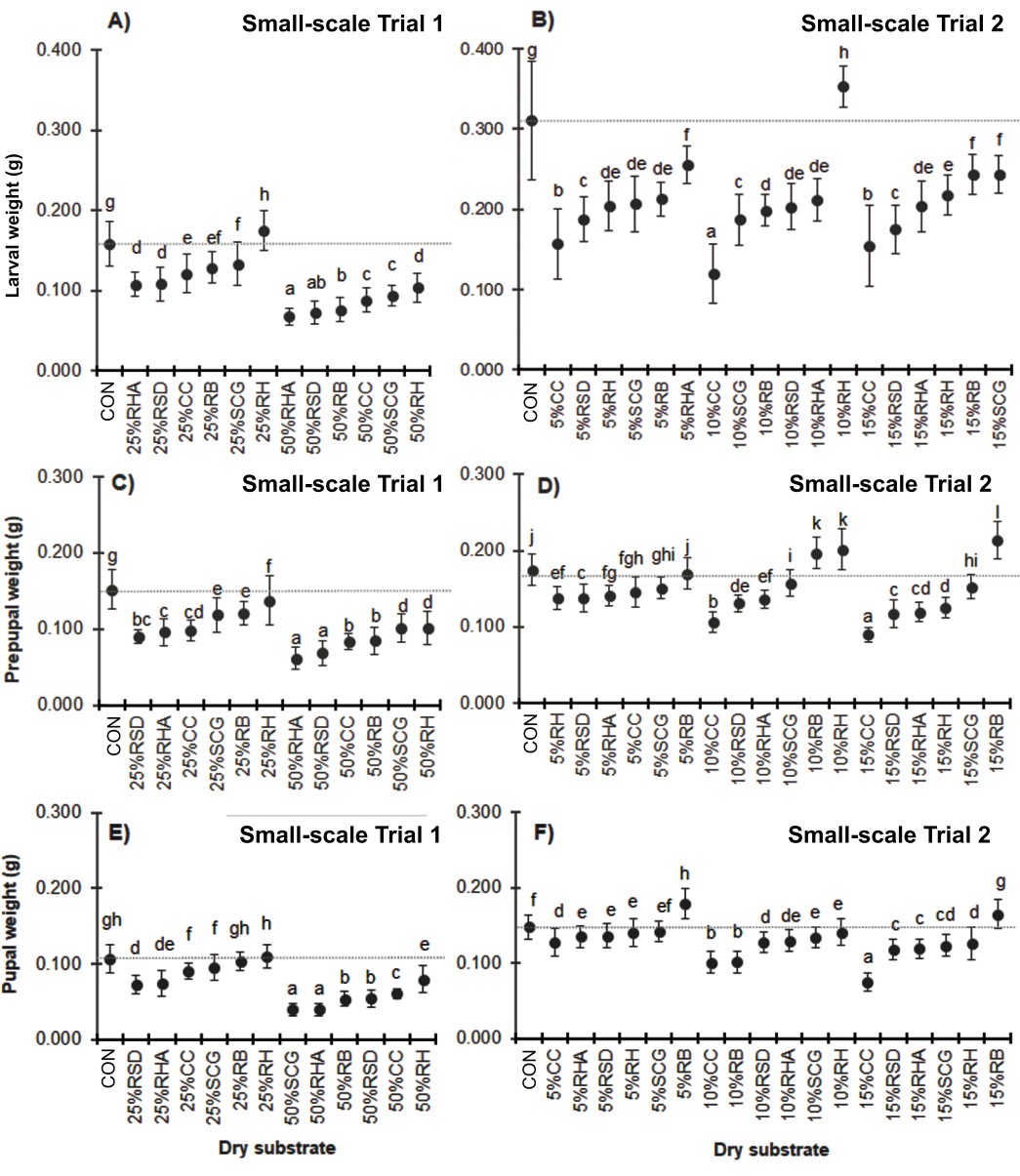

**Figure 3** Mean (± 1 SD) weight (g) of BSF (A, B) larvae, (C, D) prepupae, and (E, F) pupae when reared on media supplemented with various dry substrates in small-scale trails (A, C, E) 1 and (B, D, F) 2. Data are shown as the mean ± 1 SD, derived from four trials. Different letters indicate significant differences among substrate treatments ($P < 0.05$).

dry substrate. However, high-level mixtures resulted in a reduction in the proportion of female offspring, particularly in the case of 50% RHA ($29.1 \pm 6.2\%$) and 50% RSD ($35.4 \pm 3.0\%$; $F = 6.416$, $df = 12$, $P = 0.000$) (Fig. 4).

## Moisture removal efficiency of each dry substrate

Vegetable wastes typically have a high moisture content of $61.21 \pm 0.82\%$ (w/w), even after being minced and having their juices removed. In the control group, daily monitoring of

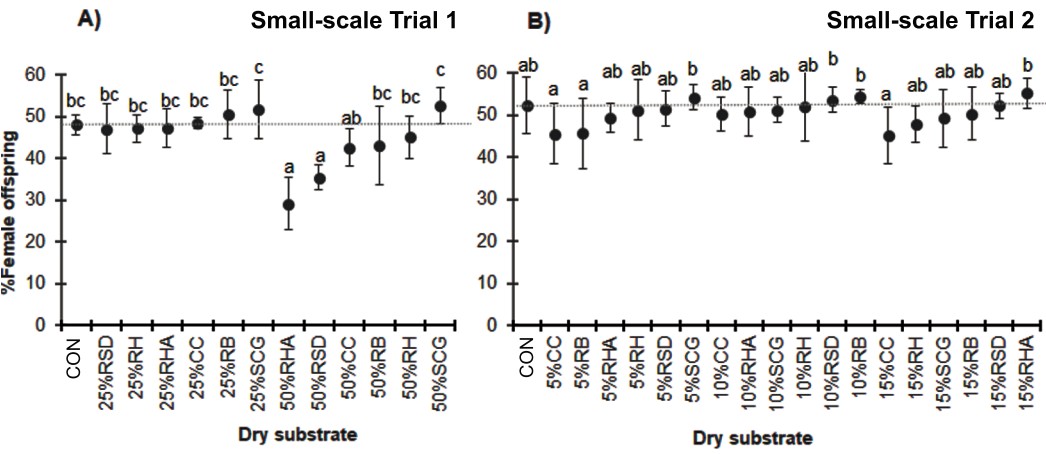

**Figure 4  Percentage of female adult BSF offspring when reared on various dry substrates in small-scale trials (A) 1 and (B) 2.** Data are shown as the mean ± 1 SD, derived from four replicates. Different letters indicate significant differences among substrate treatments ($P < 0.05$).

the substrate moisture content showed an increase starting from day 5, eventually becoming liquid during prepupal development. By mixing different dry materials in varying quantities (as shown in Table 1), the moisture level of the substrate can be controlled. The M for each dry substrate was determined by comparing the moisture content before and after the addition of the dry substrate. The M was found to be correlated with the larval duration ($r = -0.457$, $P = 0.000$). The RB exhibited the highest absorbent properties among the tested dry substrates, followed by CC and RH in that order. However, when these substrates were added in larger proportions (ranging from 25% to 50%), they caused the substrate diets to dry out, leading to elevated MRs and diminished biomass levels. Among the dry substrates in the small-scale trial 2, the use of 15% RB (with a moisture content of 59.01 ± 1.91% and an M of 10%) provided good support for BSFL compared to other dry substrates. The dry substrate with the lowest M was RHA (2.49–3.34%).

### Scale-up tests for medium-scale production

Four treatments (5% RHA, 10% RH, 5% RB, and 15% SCG) were selected from the small-scale tests for evaluation in medium-scale production using the same quantities of added dry substrate. With the exception of 15% SCG, the M of the substrates was reduced compared to that in the small-scale tests (Table 2). Consequently, the substrates did not dry out during development of the prepupae and larger larvae and pupae were produced. Figure 5 reveals that substrate mixtures containing 15% SCG and 5% RHA resulted in a significantly increased weight of the larvae and prepupae ($F = 63.220$, $df = 4$, $P = 0.000$ and $F = 137.642$, $df = 4$, $P = 0.000$, respectively).

## DISCUSSION

Globally, fruits and vegetables are common organic wastes that have the potential to be used as BSFL substrates (*Jucker et al., 2017*) with a high waste-to-biomass conversion level

**Table 1** Moisture range (%) and the percentage of moisture removal (M) (%) of various substrates in two small-scale trials.

| Treatment (% (w/w)) | Moisture range (%) | M (%) |
|---|---|---|
| **Small-scale trial 1** | | |
| CON | 60.25–63.10 | – |
| 25% RHA | 27.30–59.93 | 55.40 |
| 25% SCG | 23.63–60.28 | 61.40 |
| 25% RSD | 15.50–58.85 | 74.68 |
| 50% SCG | 8.23–59.15 | 86.56 |
| 50% RHA | 7.68–57.08 | 87.46 |
| 25% RH | 5.95–60.08 | 90.28 |
| 50% RSD | 4.20–56.23 | 93.14 |
| 50% RH | 1.08–53.53 | 98.24 |
| 25% CC | 0.95–61.85 | 98.44 |
| 50% CC | 1.00–62.93 | 98.37 |
| 50% RB | 0.23–55.43 | 99.63 |
| 25% RB | 0.00–58.70 | 100.00 |
| **Small-scale trial 2** | | |
| CON | 60.53–63.47 | – |
| 5% RHA | 60.08–62.93 | 2.49 |
| 10% RHA | 59.78–62.83 | 2.98 |
| 5% RB | 59.55–61.85 | 3.34 |
| 15% RHA | 59.55–62.25 | 3.34 |
| 5% CC | 59.55–62.73 | 3.34 |
| 5% SCG | 59.45–62.45 | 3.51 |
| 5% RH | 59.43–62.18 | 3.55 |
| 10% SCG | 59.23–61.35 | 3.87 |
| 10% CC | 59.13–61.63 | 4.03 |
| 5% RSD | 59.08–62.73 | 4.11 |
| 10% RH | 59.05–62.03 | 4.16 |
| 15% SCG | 59.00–61.58 | 4.24 |
| 10% RSD | 58.63–61.85 | 4.84 |
| 15% RSD | 58.23–61.05 | 5.49 |
| 10% RB | 58.15–61.50 | 5.62 |
| 15% RH | 57.55–61.28 | 6.59 |
| 15% CC | 56.93–59.43 | 7.60 |
| 15% RB | 55.73–62.15 | 9.55 |

(*Bekker et al., 2021*). However, their high moisture content presents a challenge in BSF production. Studies have shown that adjusting the moisture levels can both improve the survival rates and increase the obtained BSFL biomass when using certain substrates, such as apples and spinach (*Ribeiro, Costa & Ameixa, 2022*). The lower nutritional value of fruits and vegetables, in terms of their low protein concentration (around 13%) and high a carbohydrate-to-protein ratio, can lead to a decreased larval growth performance (*Bonelli et al., 2020*; *Jucker et al., 2017*; *Kinasih et al., 2018*; *Lalander et al., 2019*). As a generalist

**Table 2 Moisture range (%) and the percentage of moisture removal (M) (%) of media with various dry substrate additions in small- and medium-scale experiments.**

| Treatment (% (w/w)) | Small-scale | | Medium-scale | | Difference |
|---|---|---|---|---|---|
| | Moisture range | M (%) | Moisture range | M (%) | |
| CON | 60.53–63.47 | – | 62.03–63.63 | – | – |
| 5% RHA | 60.08–62.93 | 2.49 | 61.80–63.23 | 1.47 | decrease |
| 10% RH | 59.05–62.03 | 4.16 | 59.93–63.13 | 1.84 | decrease |
| 5% RB | 59.55–61.85 | 3.34 | 43.87–63.40 | 2.64 | decrease |
| 15% SCG | 59.00–61.58 | 4.24 | 61.30–63.47 | 4.81 | increase |

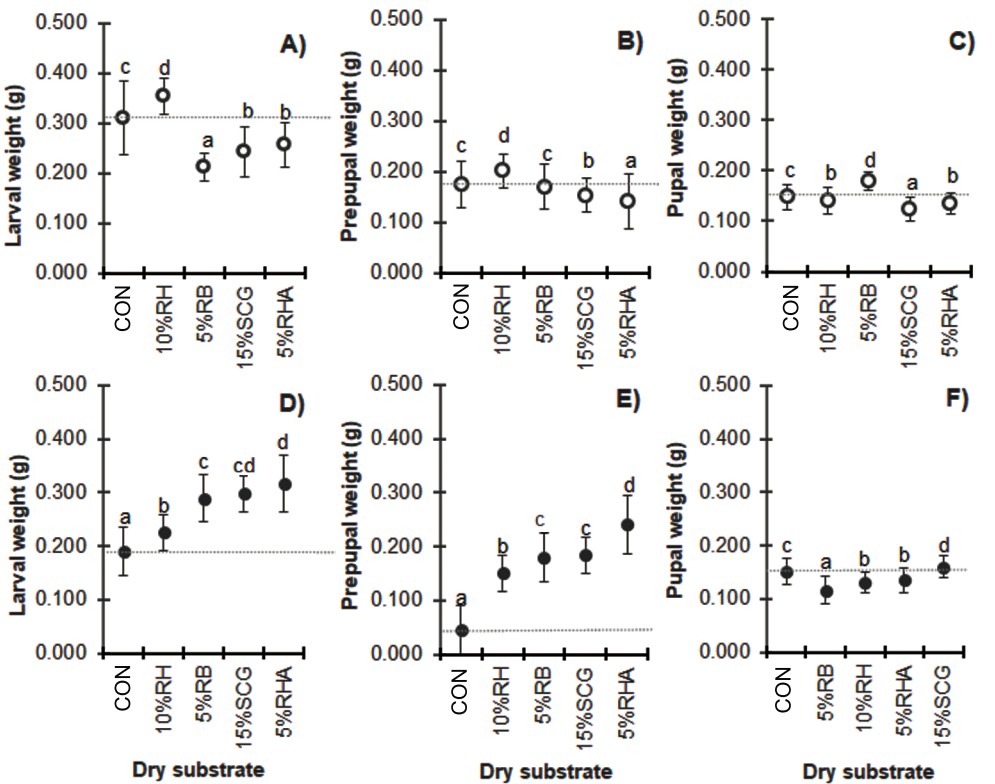

**Figure 5 Comparison of the weight (g) of BSF (A, D) larvae, (B, E) prepupae, and (C, F) pupae reared on various dry substrates in (A–C) small-scale and (D–F) medium-scale production.** Data are shown as the mean ± 1 SD, derived from four replicates. Different letters indicate significant differences among substrate treatments ($P < 0.05$).

feeder, BSFL have demonstrated their ability to adapt their digestion process and optimize larval performance (*Bonelli et al., 2020*).

Optimizing the feed composition for BSFL rearing through the incorporation of vegetables as the main ingredient can significantly improve the waste-to-biomass conversion efficiency. The combination of vegetable canteen waste and mill byproducts was previously demonstrated to augment the BSFL biomass production compared to poultry

feed despite the latter's higher protein content (*Gold et al., 2020*). Apart from the enhanced food quality, this substrate combination can also improve the moisture content. Fruits like apples, bananas, or a blend of both have a low nutrient density and a high moisture content, rendering them suitable substrates for BSFL when integrated with spent grains at a 1:1 (w/w) ratio (*Scala et al., 2020*). Wheat bran has been used to regulate the substrate moisture (*Yang & Tomberlin, 2020*).

In this investigation, RB emerged as a potentially effective dry substrate; however, numerous studies have highlighted its limitations. Previous examination of the proportion of RB mixed with coconut pulp residue revealed that BSFL did not thrive when the mixture contained over 50% (w/w) RB due to the resulting high substrate temperatures (*Muchdar et al., 2021*). Similarly, and consistent with this study, RB could sustain BSF larval growth and rapid development when used at less than 25% (w/w) in the substrate due to its elevated protein content. However, BSFL exhibited a higher MR when exposed to 25% and 50% (w/w) RB substrates since these resulted in high substrate temperatures and substrate drying out. This is consistent with the recommendation for the control of high moisture substrates of placing the RB on the substrate's outer edge (*Barrett et al., 2023*), where the small quantity of RB on the surface of the media prevents excessive substrate heating.

The utilization of toxic dry substrates deserves considerable attention. For instance, CC has an abundant tannin content with the potential to release these compounds into the BSFL substrate as they are water soluble substances. These can be toxic to the digestive tissues of insects, including BSFL, and they also possesses antimicrobial properties that can potentially adversely alter the insect's gut microbiome as well as substrate fermentation (*Barbehenn & Peter Constabel, 2011*). Despite CC exhibiting a high water-holding capacity (*Evans, Konduru & Stamps, 1996*) and its potential to effectively manage a substrate with a high moisture content (Table 1), its utilization appears to result in a high BSFL mortality and low BSF biomass. Nevertheless, this study did not specifically examine tannin or other toxic substances within the substrates. The removal of tannin through leaching by washing with 60 °C water (*Tejano, 1985*) or microwave treatment (*Adeel et al., 2020*) before adding the CC to the larval substrate for rearing BSFL could potentially enhance its beneficial properties.

Likewise, the presence of toxic phenolic compounds in SCG could impact the larval growth (*Permana & Ramadhani Eka Putra, 2018*; *Sideris et al., 2021*). However, fermented SCG, derived from cofermentation with pineapples at levels of less than 60% (w/w) in the substrate mixture gave larger BSFL (*Khaekratoke, Laksanawimol & Thancharoen, 2022*). Although the SCG used in this study underwent self-fermentation, the BSFL exhibited similar trends. Additionally, the use of SCG has been recommended for its ability to minimize the odor from substrate decomposition.

The particle sizes of dry substrates are related to the level of oxygen diffusion within the media. In conditions of high moisture, the fine particle sizes of dry substrates, like SCG, can lead to a compact texture. This can limit the level of oxygen diffusion, particularly at the lower regions of the substrates and so prevent utilization of this substrate region (*Barrett et al., 2023*). Fortunately, the substrate depth in the medium-scale study fell within a range of approximately 5–7 cm, allowing the larvae to exhibit their typical feeding behavior.

Furthermore, the tested substrates received active ventilation once a day. However, the observed differences in the BSFL growth performances between the small-scale and medium-scale studies are likely to be attributed to the differences in substrate depth.

Both RH and RHA possess the ability to absorb heavy metals (*Kumar et al., 2013*; *Lata & Samadder, 2014*; *Syuhadah & Rohasliney, 2012*). Previously, RH biochar was recommended for use when contaminated organic waste substrates were used for rearing BSFL due to its capacity to absorb up to 97% of the cadmium (*Mutanekelwa, Okumu & Andika, 2022*). However, that study did not evaluate the efficiency with live BSFL, nor did it investigate the effects of different proportions of RH biochar on the larval mortality or growth. Our data demonstrate that substrate mixtures containing RHA correspondingly reduced the pH of the substrates depending on the added quantity (pH = 6.2 for 50% RHA), whereas the pH of the RH substrate ranged from 7.0–7.1. *Ma et al. (2018)* discovered that substrates with a pH level between 6.0 and 8.0 were optimal for substantial BSFL production, in contrast to levels below pH 4.0 that were detrimental. Therefore, in this study, the observed high MRs of BSFL in substrates with low quantities of dry substrates is not likely to have been caused solely by a low pH. With respect to the %M, it is evident that substrate mixtures with low quantities of RHA (5–15%) exhibited a minimal %M, resulting in a high moisture content that led to an increased MR during the prepupal stage. In contrast, substrates containing 5–10% RHA could support a larger BSFL mass compared to those with higher quantities of RHA.

Typically, RH has a higher %M and larger particle sizes than RHA. *Yakti et al. (2023)* demonstrated that fibers with larger particle sizes possess a greater water-holding capacity and facilitate an improved aeration of larval substrates, consequently leading to larger larval sizes and a lower MR. This is consistent with the findings of the present study, where RH demonstrated the capacity to effectively support BSFL performance, even when present in significant quantities. Substrates containing 25–50% (w/w) RH could promote shorter BSFL development and yield a larger BSFL biomass compared to the control, but without a significant difference in the MR compared to the control group.

Comparison of the BSFL growth performance between the small-scale and medium-scale experiments revealed distinct outcomes. Evidently, in the medium-scale trials, substrates with dry substrate mixtures yielded a greater BSFL biomass, particularly in the cases of 5% RB, 15% SCG, and 5% RHA. Notably, factors such as the container volume, substrate surface area, and substrate thickness likely contributed significantly to these differences. Nevertheless, the %M of most substrates was decreased in the medium-scale trials, making them suboptimal for the pupation of prepupae. As a result, some prepupae exhibited a climbing behavior to escape from the damp substrate. In light of this, we propose considering the incorporation of higher quantities of the dry substrate when working with larger-scale setups.

Based on the findings of this investigation, the selection of dry substrates for controlling the substrate moisture level requires careful consideration of diverse properties, including the water-holding capacity, nutritional content, presence of toxic compounds, particle sizes, and substrate cost. We have detailed the pros and cons of each dry substrate for moisture control in Table 3, where RB and RH emerge as promising dry substrates for

**Table 3  Pros and cons of each dry substrate supplement for moisture control in BSF rearing.**

| Dry substrate | M (%) | MR (%) | Larval duration(d) | BSFL growth (g) | Recommended quantity (% (w/w)) | Other properties |
|---|---|---|---|---|---|---|
| RB | 3–100 | 5.2–25.3 | 14–15 | 0.076–0.243 | ≤15% | High nutritional content, avoid substrate dry out, high temperature substrate. |
| RH | 4–98 | 4.2–11.4 | 13–16 | 0.104–0.243 | up to 50% | Ability to absorb heavy metals. |
| RHA | 3–87 | 3.6–16.7 | 13–17 | 0.068–0.255 | ≤15% | Slightly acidic. |
| CC | 3–98 | 4.5–19.8 | 14–16 | 0.088–0.156 | n/a | Remove toxic substance before uses. |
| SCG | 4–87 | 6.2–18.3 | 13–17 | 0.094–0.243 | ≤25% | Minimize the odor. |
| RSD | 4–93 | 4.6–8.8 | 14–16 | 0.073–0.203 | ≤10% | – |

use in BSF rearing. A quantity of RB less than 15% effectively controls substrate moisture content and promotes BSFL growth with a shorter harvest time. BSFL grown in substrates with less than 50% RH exhibit low MR and large larval growth. However, it is worth mentioning that a combination of different dry substrates can be employed to strike a balance in substrate moisture and enhance the efficiency of vegetable-fruit substrates for successful BSFL rearing.

# CONCLUSIONS

Several low-cost agricultural byproducts exhibit the potential to serve as supplements for BSFL rearing substrates to effectively manage the substrate moisture level. Based on their moisture control efficacy and impact on BSFL performance, RB and RH are potential dry substrates for BSF rearing. However, care in the use of RB is required to prevent its excessive usage (>15%), which could lead to increased temperatures or substrate desiccation. In contrast, CC stands out as an ideal option for moisture control, yet a prerequisite for its application involves the removal of tannins. The combination of substrates can be performed to enhance the efficiency and cost-effectiveness.

# ACKNOWLEDGEMENTS

The authors thank the Royal Project Foundation for providing the vegetable waste, singularly used in the experiment to raise BSF stock. We also thank Miss Kanyanat Khaekratoke for helping us in the lab.

## Funding

This study was financed by the Kasetsart University Research and Development Institute (KURDI) (Grant No. FF(KU) 14.64). The funders had no role in study design, data collection and analysis, decision to publish, or preparation of the manuscript.

## Grant Disclosures

The following grant information was disclosed by the authors:

Kasetsart University Research and Development Institute: FF(KU) 14.64.

## Competing Interests

The authors declare there are no competing interests.

## Author Contributions

- Parichart Laksanawimol conceived and designed the experiments, performed the experiments, analyzed the data, prepared figures and/or tables, authored or reviewed drafts of the article, and approved the final draft.
- Pritsana Anukun performed the experiments, analyzed the data, prepared figures and/or tables, and approved the final draft.
- Anchana Thancharoen conceived and designed the experiments, analyzed the data, prepared figures and/or tables, authored or reviewed drafts of the article, equipment and materials in laboratory, and approved the final draft.

## Data Availability

The raw measurements are available in the Supplementary Files.

## Supplemental Information

Supplemental information for this article can be found online at http://dx.doi.org/10.7717/peerj.17129#supplemental-information.

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
