# Peer review of "Use of different dry materials to control the moisture in a black soldier fly (Hermetia illucens) rearing substrate"

_PeerJ, doi:10.7717/peerj.17129_

## Round 0.1 · original submission · Major Revisions

Both reviewers and I saw merit in your manuscript and are keen to see it progress. Further information is required in regard to the experimental design and methodology - if you have information on water-holding capacity and nutritional profile can you please add this information?

Reviewer 1 ·

Basic reporting

Clear and unambiguous, professional English used throughout.

Experimental design

Experimental design is not clearly explained and consequently must be improved.

Authors should add references to all procedures used in the study.

Authors should equation numbers to all equations.

Authors wrote ‘In the large-scale experiments, a plastic tray with dimensions of 60 cm (L) x 40 cm (W) x15 cm (H) was used’. 60 cm x 40 cm x 15 cm cannot be considered as large scale. Authors should replace the term large scale with a more appropriate term.

Validity of the findings

In conclusion, authors wrote that ‘Each dry substrate boasts distinct physical and chemical attributes, encompassing a different water-holding capacity, nutritional profile, presence of potentially toxic compounds, particle sizes, etc’. However, no data on water-holding capacity and nutritional profile were investigated in this study. Authors should make conclusions based on the data obtained in this study.

The meaning of asterisk symbol (*, **, ***, **** and *****) in Table 3 is not clear. Authors should explain the meaning of each symbol.

Authors should clearly explain the rationale of recommended quantity in Table 3.

Average and range values in Table 1 and Table 2 are redundant as readers can determine the average values from the provided range values.

Additional comments

Np additional comments

Reviewer 2 ·

Basic reporting

The paper addresses the use of various substrates to control the humidity and moisture environment in which the black soldier flies (BSF) are reared. The study is important and provides a useful guide to those wishing to establish BSF farms themselves.

I found this manuscript to be very professionally written in terms of english, references and layout. Very well done.

Experimental design

The experimental design is sound and demonstrates the iterative nature in which designs evolve.

Validity of the findings

The findings are sound with certain substrates proving to be much more effective than others as well as additional results were obtained on the useful or less useful side properties of the substrates, such as metal adsorption and prior cleaning of substrates may be required.

Additional comments

Q1. Table 1. Why are the SDs of the various dry substrates orders of magnitude larger than the SD of the control in the small scale trial 1 please?
Table 1, please explain Em % in the caption so reader doesn't have to go back to text. (moisture removal).

Conclusions, line 388. Pls remove etc. Pls list all items being referred to. Perhaps state this is not an exhaustive list.

---

## Round 0.2 · Minor Revisions

Both reviewers are happy with the changes you have made to enhance the clarity of the manuscript. Reviewer 2 would like some further clarification made before the paper is accepted. I agree that these issues should be rectified inthe updated manuscript.

Thanks for your work on the manuscript

Reviewer 1 ·

Basic reporting

No comment

Experimental design

No comment

Validity of the findings

No comment

Additional comments

Authors have revised the manuscript based on the input from reviewers. Hence, the manuscript can be accepted for publication.

Reviewer 2 ·

Basic reporting

I have gone through an English and grammar editing process, I've put tracked changes and highlighted some of the changes I've made here, more are in the attached track changed manuscript.


- [ ] English language check

‘Byproduct’ consistent spelling

Pg4 Line 17, removed ‘the’

Pg 4, Line 87, removed ‘the’

Pg4 Line 99, content

Experimental design

I've focussed on the requests of the editor.

Validity of the findings

- [x] Double check figures and tables

Figure 2, given the size of the standard error bars, I’m struggling to see how treatments b and c are meant to be significantly different from one another given their individual b and c letter designations. Could this be justified please?

The same query arises in Figure 3.

Likewise figure 5.



- [x] Open Raw Data

The first sheet tab, Larval duration_small appears unusual, are these lines in the excel sheet individual larvae?
There appears to be a graph and an orange highlights in the data placed randomly. (Row 385 and row 5069). What do these mean?
Larval Growth Small_scale data looks reasonable.

Additional comments

n/a

Annotated reviews are not available for download in order to protect the identity of reviewers who chose to remain anonymous.

---

## Round 0.3 · accepted · Accept

Thank you for making all the minor editorial changes requested in the previous rounds of review. Your manuscript is now acceptable for publication and will make a great contribution to the Black Soldier Fly literature.